# Brave New World of Artificial Intelligence: Its Use in Antimicrobial Stewardship—A Systematic Review

**DOI:** 10.3390/antibiotics13040307

**Published:** 2024-03-28

**Authors:** Rafaela Pinto-de-Sá, Bernardo Sousa-Pinto, Sofia Costa-de-Oliveira

**Affiliations:** 1Division of Microbiology, Department of Pathology, Faculty of Medicine, University of Porto, Alameda Prof. Hernâni Monteiro, 4200-319 Porto, Portugal; up201807421@edu.med.up.pt; 2Department of Community Medicine, Information and Health Decision Sciences, Faculty of Medicine, University of Porto, 4200-319 Porto, Portugal; bernardo@med.up.pt; 3Center for Health Technology and Services Research—CINTESIS@RISE, Faculty of Medicine, University of Porto, 4200-319 Porto, Portugal

**Keywords:** artificial intelligence, machine learning, antimicrobial stewardship, antimicrobial resistance

## Abstract

Antimicrobial resistance (AMR) is a growing public health problem in the One Health dimension. Artificial intelligence (AI) is emerging in healthcare, since it is helpful to deal with large amounts of data and as a prediction tool. This systematic review explores the use of AI in antimicrobial stewardship programs (ASPs) and summarizes the predictive performance of machine learning (ML) algorithms, compared with clinical decisions, in inpatients and outpatients who need antimicrobial prescriptions. This review includes eighteen observational studies from PubMed, Scopus, and Web of Science. The exclusion criteria comprised studies conducted only in vitro, not addressing infectious diseases, or not referencing the use of AI models as predictors. Data such as study type, year of publication, number of patients, study objective, ML algorithms used, features, and predictors were extracted from the included publications. All studies concluded that ML algorithms were useful to assist antimicrobial stewardship teams in multiple tasks such as identifying inappropriate prescribing practices, choosing the appropriate antibiotic therapy, or predicting AMR. The most extracted performance metric was AUC, which ranged from 0.64 to 0.992. Despite the risks and ethical concerns that AI raises, it can play a positive and promising role in ASP.

## 1. Introduction

One Health is, according to the One Health High-Level Expert Panel, “an integrated, unifying approach that aims to sustainably balance and optimise the health of people, animals and ecosystems” [1]. This inextricable link between these actors applies to various fields of health and, inherently, to the growth of antimicrobial resistance (AMR).

AMR is a growing public health problem due to its effect in reducing the effectiveness of antimicrobial therapy and increasing the severity, incidence, and cost of infection [2]. AMR’s emergence, evolution, and spread stem from (i) the widespread and inadequate antimicrobial use in animals and clinical practice, (ii) contaminated environments, (iii) and insufficient infection control measures [3]. This increases the threat of the emergence of super-resistant bacteria [4]. The rapid development and dissemination of the mechanisms of resistance through antibiotic resistance genes (ARGs) to antibiotics used in the clinical setting, adding to the slow and infrequent access to new antimicrobials in recent years, makes AMR one of the most severe threats to global public health in the 21st century. 

AMR levels are detected by antimicrobial susceptibility testing (AST). However, this method involves culture of the microorganisms, which can take 2–5 days. This delay in the prescription of the most effective antimicrobials leads to the prolongation of empiric therapy, contributing to the rise of AMR, so measures must be taken to combat this, including improved communication and education about the topic, adequate hygiene for infection control, surveillance practices, antimicrobial stewardship, swifter methods for AMR identification, and the use vaccines and bacteriophages [2,3]. 

Antimicrobial stewardship programs (ASPs) are a set of interventions aimed at optimizing the use of antimicrobials and, therefore, reducing costs, improving therapeutic outcomes, and reducing AMR [5]. ASPs were introduced in 1974 by McGowan and Finland [6], are applied to human healthcare, animal health, and the environment, and involve the optimal selection, dosage, and duration of therapy as well as the control of its use, which can be achieved with programs that recommend the appropriate adjustments. Typically, an ASP may involve pharmacists and infectious diseases physicians, and the tools available for these teams include limiting formularies, restricting certain classes of antimicrobials, cycling of antibiotics, decision support, and staff education about the optimal antimicrobial considering the patient [5]. These interventions are primarily used in hospital settings such as in intensive care units (ICUs), pediatrics, and neutropenic patients [7,8,9]. Still, efforts should be made for their application in outpatient settings to achieve a significant impact on the reduction of AMR [10]. The measurement of the impacts of ASPs can be categorized into antibiotic use, process and quality measures, costs, and clinical outcome measures, with the latter being the most relevant focus in practice [11]. There are challenges in implementing ASPs, including a lack of motivation for change and awareness, a lack of oversight and control of antimicrobial use in many countries, and over-the-counter therapy [12].

Artificial intelligence (AI) began developing in the 1950s, and its first use in healthcare was in the form of expert systems, which were based on rules provided by medical experts, but were never applied in practice [13]. Machine learning (ML) was developed to overcome the limitation of expert systems that need a large number of rules captured, since ML can find new rules from the data provided, based on their quality and volume [13], benefitting mainly from the enormous amount of health data gathered after the implementation of electronic health records. As some real patient situations are more complex and heterogeneous than a single guideline or the experience of an expert, ML can be a tool used to help decision-making in these situations, since it can analyze a great number of electronic records in a way similar to experts’ logical deduction. ML algorithms can be supervised or unsupervised, and some examples include support vector machines, artificial neural networks, random forests, decision trees, and logistic regression [10,14]. Previous studies have shown that this technology has been used in numerous healthcare fields, including infectious diseases [13]. It has been proved to be useful in prediction [15] and early detection [16] of sepsis, diagnosis of infection [17], prediction of treatment success [18], prediction of antimicrobial resistance [19], and treatment selection [20], meaning that it may be an effective tool to put into practice in antimicrobial stewardship teams, bettering their programs.

This systematic review aims to explore the use of AI in ASPs and summarizes the predictive performance of ML algorithms used in antimicrobial stewardship, compared with clinical and antimicrobial stewardship teams’ decisions, in inpatients and outpatients who need antimicrobial prescription. Studies were selected and screened from January 2010 until December 2022 in the electronic bibliographic databases of PubMed, Scopus, and Web of Science by using a combination of terms such as artificial intelligence, antimicrobial resistance, and stewardship. The protocol of this review was registered in the PROSPERO database (CRD42023470594). 

## 2. Results

### 2.1. Characteristics of the Included Studies

A total of 4658 citations were identified from the three databases and, after removing the duplicates, 2839 were eligible for screening. A total of 1086 articles were assessed for eligibility and eighteen [20,21,22,23,24,25,26,27,28,29,30,31,32,33,34,35,36,37] were included in this systematic review (Figure 1). Most studies were excluded because they did not study the application of machine learning models nor their predictive performance or because they were not applied to hospital inpatients and outpatients with infections, such as studies in vitro or regarding drug development.

Characteristics of the eighteen included studies are available in Table 1. All the studies were published since 2016 and in English. One of the studies is an abstract presentation at a congress in video format [37]. One of the studies was from a low-/middle-income country [36], with the rest being from high-income countries. The number of features included in the machine learning algorithms ranged from 6 to 788. The patients included were from different settings; one (5.5%) study was designed for outpatients [35], and two were only applied to ICU patients [26,29]. The number of patients ranged from 48 (on a validation set) to 382,943. Two [20,34] of the studies had a prospective design, with the remaining being retrospective observational studies.

The most common ML algorithms used were logistic regression (12.1%), random forest (12.1%), support vector machine (7.6%), and k-nearest neighbors (6.1%) (Figure 2). The measurements used for predictive performance were not consistent between different studies, but the area under the curve (AUC) (15.9%), sensitivity (9.1%), specificity (8.0%), and precision (6.8%) were the most regularly used (Figure 3).

The features included in the algorithms were divided into the following groups: demographics, adult patients, pediatric patients, clinical, laboratory/microbiological, comorbidities, type of infection, and ICU. The most used features were demographical followed by laboratory/microbiological. Information about the features used in each study is available in Table 2.

The most common validation method was k-fold cross-validation (fivefold and tenfold) to avoid overfitting. Not all included studies provided information about handling missing data or methods to avoid overfitting, and two studies did not reference the model validation method [20,37]. Corbin C.K. et al. [22] replicated the process on an external validation cohort in Boston.

### 2.2. Risk of Bias/Quality Assessment

All the studies were rated as being of “fair quality” by the NIH Quality Assessment Tool for Observational Cohort and Cross-Sectional Studies; fourteen studies were rated as 57.1% and four [22,26,32,35] were rated as 64.3%. The participation rate, variation in amount or level of exposure, and loss to follow-up criteria were not applied to any of the studies. Only one study [35] provided a sample size justification or power description. No study reported information about blinding the assessors, and only three studies [22,26,32] met the criterion on the statistical adjustments of potential confounding variables. The answer to each of the fourteen criteria, as well as the quality rating, are available in Table 3. 

The risk of bias (ROB) and the applicability for model prediction of the eighteen included studies were also assessed by PROBAST (Table 4). Only two studies were ranked as being of “low concern” in the analysis domain [22,24]; six studies were defined as being of “unclear concern” [20,21,23,25,26,29], and ten were ranked as being of “high concern” [27,28,30,31,32,33,34,35,36,37]. In these studies, no information was provided regarding how missing data had been handled. Overall, only one study was rated as having a low ROB [24]. Regarding applicability, one study ranked as being of “high concern” and as having a high ROB due to the lack of participant information and lack of definition of the inclusion and exclusion criteria [27].

### 2.3. Predictive Performance of Artificial Intelligence Algorithms

The most evaluated performance metric was AUC, which ranged from 0.64 to 0.992 (the highest value was obtained by the multilayer perceptron). This algorithm also achieved the highest sensitivity (0.967) and specificity (0.992) for auditing appropriate surgical antimicrobial prophylaxis. The highest precision was achieved by the gradient boosted tree, with an average precision of 0.99 for the selection of vancomycin + meropenem. The other main results are available in Table 1. All the studies concluded that ML algorithms were useful to assist antimicrobial stewardship teams in multiple tasks such as identifying inappropriate prescribing practices [20], choosing the appropriate antibiotic therapy [22,23,34,36], auditing surgical antimicrobial prophylaxis [24], predicting personal risk of treatment-induced emergence of resistance [25], estimating patient outcomes under the contrasting scenarios of stopping or continuing antibiotic treatment [26], predicting AMR [27], and identifying patients at low risk of bacterial infections [29].

Regarding the choice of the most appropriate antibiotic therapy, the model with the best performance was random forest, with an area under the curve of 0.80 (95% CI 0.66–0.94) for the prediction of susceptibility to ceftriaxone, 0.74 (0.59–0.89) for ampicillin and gentamicin, and 0.85 (0.70–1.00) for susceptibility to neither [36].

For the identification of inappropriate prescribing practices of piperacillin-tazobactam, the algorithm applied was the supervised learning module of APSS (antimicrobial prescription surveillance system). It obtained an overall positive predictive value of 74% (95% CI, 68–79), with sensitivity (recall) of 96% (92–98) and accuracy of 79% (74–83) [20].

Logistic regression achieved a 67% reduction in second-line antibiotics relative to clinicians and an 18% reduction in inappropriate antibiotic therapy [35].

## 3. Discussion

### 3.1. Main Findings

A systematic review of the utility of AI in antimicrobial stewardship for inpatients and outpatients who needed antimicrobial decisions was conducted, and eighteen studies were included. Logistic regression and random forest were the most used algorithms. AUC was the most common predictive performance measure, and the highest value was obtained by the multilayer perceptron [24]. The most studied application of AI in ASPs was the use of AI for choosing the appropriate antibiotic therapy. In one study, the algorithm used was a semi-supervised decision support system [21]; the remaining algorithms applied supervised ML algorithms, which are generally used to make predictions. All the studies concluded that AI algorithms can help choose the best antimicrobial therapy, benefiting, for example, the control of AMR. These results are aligned with what has been found about AI use in infectious diseases, since other systematic reviews summarize its applicability in antimicrobial susceptibility testing [14], predicting antimicrobial resistance [38], prediction of treatment success, diagnosis of infection, and prediction of sepsis [13].

### 3.2. AI and Antimicrobial Stewardship

Although AI can be helpful in addressing the large amount of data gathered nowadays and performing repetitive tasks, there are some risks and ethical concerns that must be considered, for example, the possibility of the algorithm making associations between features and outcomes that are not relevant or are without physiological/clinical rationale, the blind obedience/overdependence on AI, liability, or accountability in case of mistakes [39]. Clinical decisions are complex and include factors about the patient, the disease, the economy, or the environment, so the algorithm should not uniquely make the final decision. “Black box” is an aspect of AI that raises concerns, since these algorithms cannot explain the underlying mechanism to generate outputs, and we may not know the source of data input. This has a significant impact on transparency and trust [40,41]. In response to the rise of AI health technologies, the WHO published six regulatory areas of AI for health, including the transparency of development processes, external data validation, cybersecurity, and data protection [42]. The WHO emphasizes the need for collaboration between regulators, patients, healthcare professionals, industry, and governments to ensure the compliance of AI models with regulation. The application of AI on antimicrobial stewardship programs is still very limited, as seen in the few studies included in this systematic review. The methodological heterogeneity and the reduced number of diseases in which AI has been applied on ASPs restrict the widespread use of ML models in antimicrobial stewardship. Tools based on AI for this purpose are still in a development phase before they can be safely implemented in healthcare. 

Addressing the perception among some clinicians that the use of AI in antimicrobial stewardship is more of a mirage than a reality necessitates a clear discussion on its evident benefits. Implementing AI requires a calculated investment in technology and skilled data analysts, with the scale dependent on each hospital’s needs. A thorough cost/benefit analysis is vital, showcasing the expenses and expected advancements in healthcare efficiency and patient care quality. Embracing AI, despite initial doubts, is crucial for the evolution of antimicrobial stewardship, moving the perception of antimicrobial stewardship from skepticism to accepted implementation.

### 3.3. Limitations of the Studies Included

The research on AI applications in ASP is mostly from high-income countries, which can introduce bias on the algorithms and inequalities in healthcare because it does not represent the entire population [43]. This may happen because low- and middle-income countries may face more challenges to implement systems allowing for the collection of large amounts of structured health data, access to health is scarcer, and the financial support for implementing AI algorithms needs to be improved. Efforts should be made to include data from these populations in training and validation datasets.

There needs to be a publicly recognized tool for quality and risk assessment of ML prediction models. PROBAST and the National Institute of Health (NIH) Quality Assessment Tool for Observational Cohort and Cross-Sectional Studies were used for a more complete assessment. For most studies, there was a lack of information about sample size justification or power description and a poor description of the statistical adjustment of confounders. This is a concern, since AI algorithms can provide biased results if the information input is subject to uncontrolled biases. Bolton et al. [26] consider that the models may have “learned” the association between less severe patients receiving fewer antibiotics and, therefore, having a shorter ICU length of stay, causing some confounding. PROBAST assessment of ROB raises concerns, since only one study was ranked as “low concern”. This is mainly due to the analysis domain, as not all the included studies provided information about the handling of missing data or the methods to avoid overfitting, and two studies [20,37] did not report information on validation methods. One of the studies performed external validation of the model [22], which raises concerns about the generalizability of the algorithms used in the other studies. G. Eickelberg and colleagues [29] state that their future research will focus on external validation and clinical utility assessment of the models. The lack of participant information and the definition of the inclusion and exclusion criteria also raise concerns about the applicability and biases of the study’s conclusions [27]. It is relevant to note that providing participants’ information can minimize or highlight biases that can influence the application of the algorithms in specific populations in which they were not studied. Kanjilal et al. [35] admit this limitation in their study. The features selected for the ML algorithms were adequate, since they gather information that influences therapy decisions and patients’ outcomes, mainly raising low concerns. Studies on AI use in health should provide all the features included so there is more transparency and understanding of the processes involved. This will allow for analysis of whether the features have a medical reasoning behind the clinical outcome.

### 3.4. Limitations of the Review

There are some limitations to this review. The literature search was limited to PubMed, Web of Science, and Scopus articles, with no other bibliographic databases having been searched. Although this is a recent research topic, this information can be quickly complemented with more recent data. 

Publication bias is a possible limitation of this review, since it is likely that the studies with more favorable results have higher chances of being accepted for publication. Due to the diversity of the included studies (including differences in outcomes, assessed features, and the algorithms used), we could not perform meta-analysis.

It must be kept in mind that the AI algorithms are not implemented to substitute the healthcare professionals who make up antimicrobial stewardship teams but rather to assist in decision-making, mainly when a considerable amount of health data are gathered every day.

In the future, it would be interesting to research the integration of AI in ASPs, its adoption by healthcare professionals, usability and applicability, and their knowledge about the potential of using AI as a tool [44].

## 4. Materials and Methods

The systematic review was carried out in accordance with the *Cochrane Handbook for Systematic Reviews of Interventions* [45]; in addition, we followed the Preferred Reporting Items for Systematic Reviews and Meta-Analyses (PRISMA) [46] checklist for the review (Appendix A). 

### 4.1. Data Source and Search Strategy

The electronic bibliographic databases of PubMed, Scopus, and Web of Science were searched using a combination of MeSH terms and/or keywords regarding broad domains such as artificial intelligence, antimicrobial resistance, and stewardship. For this search strategy, the following query was used: (“artificial intelligence” OR “machine learning” OR “deep learning”) AND (“antibiotic resistance” OR “antibiotic resistant” OR “antifungal resistance” OR “antifungal resistant” OR “antimicrobial resistance” OR “antimicrobial resistant” OR “antibiotic susceptibility” OR “antifungal susceptibility” OR “antimicrobial susceptibility” OR “drug resistance” OR “drug resistant”). Additionally, and to avoid any bibliography loss, the terms (“artificial intelligence” OR “machine learning” OR “deep learning”) AND (stewardship) were included. Studies were selected and screened from January 2010 until December 2022, when the search results were last consulted. The search included all publication types except reviews or systematic reviews, and no language restrictions were applied.

### 4.2. Eligibility Criteria

Studies were included in this review if they assessed the performance of artificial intelligence models in ASP applied to hospital inpatients and outpatients with infections that needed antimicrobial treatment. We excluded (1) studies conducted only in vitro; (2) studies addressing non-infectious diseases such as cancer, epilepsy, or other neurologic diseases; (3) studies addressing the application of AI in food or animal production, drug development, disease diagnostic or survival or studies focusing on HIV, parasitic diseases, or tuberculosis; and (4) studies not focusing on bacterial infections.

This review intended to study the performance of AI algorithms for antimicrobial stewardship. The question being addressed can be expressed as follows:

P: Inpatients and outpatients who need an antimicrobial prescription;

I: Machine learning models used in antimicrobial stewardship;

C: Clinical or antimicrobial stewardship teams’ decision;

O: Predictive performance of ML algorithms (area under the curve (AUC), sensitivity, specificity, positive predictive value (PPV), negative predictive value (NPV), etc.).

### 4.3. Data Extraction and Synthesis

The extracted studies were uploaded to EndNote^TM^20 and Rayyan software [47] for duplicate removal, quality assessment, and further selection. Studies were selected first by title and abstract screening and then by full text reading. Both processes were independently performed by two reviewers (RPS and SCO) in a blinded, standardized manner. Eighteen studies were included in the systematic review (Figure 1).

A form was developed to extract the data from the included studies uniformly and consistently. We retrieved data on the study type, year of publication, country, study time frame, target population (demographic data), number of patients, hospital setting, type of infection, study objective, ML algorithms used, training data sets, number of features, data source (clinical and/or laboratory data), predictors, performance validation and metrics (AUC, sensitivity, specificity, etc.), and clinical outcome. Two authors (RPS and SCO) extracted data from primary studies independently.

### 4.4. Risk of Bias (ROB) Assessment

To evaluate the risk of bias of the studies included in this review, the National Institute of Health (NIH) Quality Assessment Tool for Observational Cohort and Cross-Sectional Studies and PROBAST (a tool to assess the risk of bias and applicability of prediction model studies) were used [48,49,50]. A three-point scale was used to grade the potential source of bias as good, fair, or poor. Regarding PROBAST, the risk of bias and applicability were assessed focusing on four domains (participants, predictors, outcomes, and analysis), which were evaluated for each included study. The risk of bias was defined as “high risk/concern” if the item’s answer was “No” or “Probably no” and “Unclear risk” if relevant information was absent. No studies were excluded based on quality. ROB assessment was performed independently by all authors.

### 4.5. Data Analysis

The predictive performance of the AI algorithms was extracted as some of these metrics: area under the curve, specificity, sensitivity, precision, accuracy (Table 1).

A meta-analysis was not conducted, due to the heterogeneity between the populations, algorithms, features, and aim of the studies included.

## 5. Conclusions

This systematic review focuses on various tasks where AI can be a supplemental tool for antimicrobial stewardship teams, benefiting the patient and the healthcare providers. It can assist in the identification of inappropriate prescriptions, the choice of appropriate antibiotic therapy, or the estimation of patient outcomes. This is essential in the One Health dimension, because preventing AMR and multiresistant microorganisms in humans interdependently benefits the health of animals, plants, and ecosystems. The supervised machine learning module of antimicrobial prescription surveillance systems and random forest could be useful tools for guiding the most appropriate antibiotic therapy. AI can assist antimicrobial stewardship teams, aiming at better control of AMR; thus, AI can be a valuable tool against this growing global health issue.

## Figures and Tables

**Figure 1 antibiotics-13-00307-f001:**
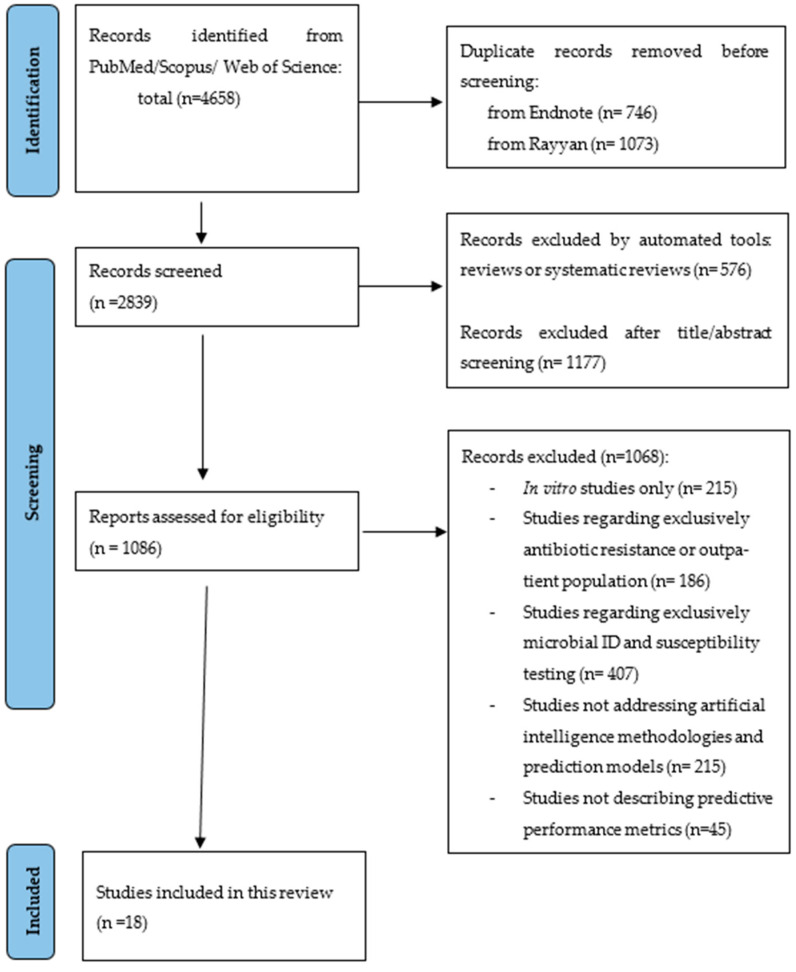
PRISMA flowchart representing the systematic search of the relevant studies.

**Figure 2 antibiotics-13-00307-f002:**
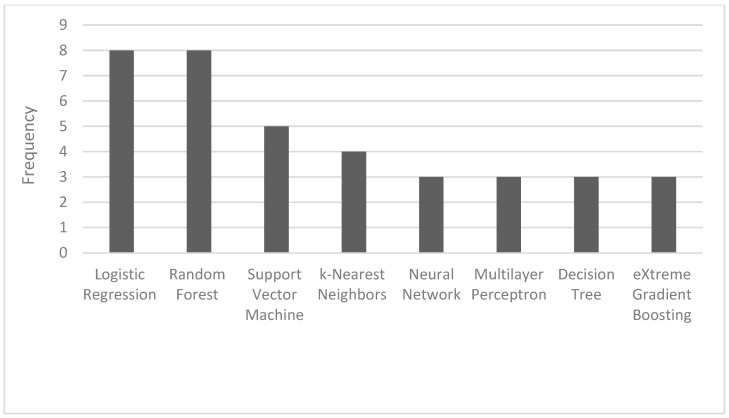
Frequency of the most used AI algorithms.

**Figure 3 antibiotics-13-00307-f003:**
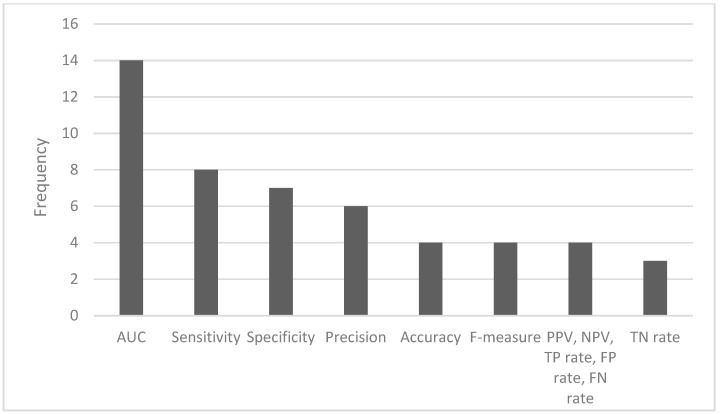
Frequency of the most used performance metrics (AUC—area under the curve; PPV—positive predictive value; NPV—negative predictive value; TP—true positive; FP—false positive; FN—false negative; TN—true negative).

**Table 1 antibiotics-13-00307-t001:** Characteristics of the included studies.

Study	Year of Publication	Country	No. Centers	Study Time Frame	Target Population	No. Patients	Infection Site	No. Features	Objective	Algorithm	Performance Measurement	Main Results
[20]	2016	Canada	1	February to November 2012	Patients monitored by APSS who received at least one prescription of piperacillin–tazobactam at the Centre Hospitalier Universitaire de Sherbrooke	421 hospitalizations	Not specified	Not specified	To evaluate the ability of the algorithm to discover rules for identifying inappropriate prescriptions of piperacillin-tazobactam	Supervised learning module of APSS, temporal induction of classification models algorithm	PPV, sensitivity, accuracy, precision	The combined system achieved an overall PPV (precision) of identifying confirmed inappropriate prescriptions of 74% (95% CI, 68–79), with sensitivity (recall) of 96% (95% CI, 92–98), and accuracy of 79% (95% CI, 74–83).
[21]	2022	Netherlands	1	January 2017–December 2018	Inpatients of the UMC Utrecht	906 cultures from 810 patients	UTI	36	To report on the design and evaluation of a CDSS to predict UTI before the urine culture results are available	CDSS using the RESSEL method; supervised models implemented in the Scikit-learn package: LR, SVM, RF, XGB and k-NN	Accuracy, sensitivity, specificity, PPV, NPV, AUC, Nneg, Npos	The predictive performance of the best-performing semi-supervised model (RF enhanced with RESSEL) had an accuracy of 76.77 (±0.97), sensitivity of 81.28 (±1.16), specificity of 70.75 (±1.85), and AUC of 80.02 (±1.00).
[22]	2022	USA	5	Stanford hospitals: January 2009–December 2019; Boston hospitals: 2007–2016	Patients who presented to Stanford emergency departments, Massachusetts General Hospital, and Brigham and Women’s Hospital in Boston	Stanford: N = 8342 infections from 6920 adult patients. Boston: N = 15,806 uncomplicated urinary tract infections from 13,862 unique female patients. Our dataset is split by time into training, validation, and test sets containing Ntrain = 5804 patient infections from 2009 to 2017, Nval = 1218 patient infections from 2018, and Ntest = 1320 patient infections from 2019.	Stanford: unspecified infection; Boston: UTI	Boston: The total number of features used in this portion of the analysis was 788. Stanford: In total, the sparse feature matrix contained 43,220 columns.	To investigate the utility of ML-based clinical decision support for antibiotic prescribing stewardship.	LR, RF, gradient boosted tree, lasso, ridge	AUROC, prevalence, average precision, antibiogram coverage rate	Stanford dataset: personalized antibiograms reallocate clinician antibiotic selections with a coverage rate of 85.9%, similar to clinician performance (84.3% *p* = 0.11). The best model class for selection of vancomycin+meropenem was gradient boosted tree, with average precision of 0.99 [0.99, 0.99] and AUROC of 0.73 [0.65, 0.81]. Boston dataset: personalized antibiograms coverage rate of 90.4%, a significant improvement over clinicians (88.1% *p* < 0.0001). The best model class for the selection of levofloxacin was LASSO, with an average precision of 0.96 [0.95, 0.96] and AUROC of 0.64 [0.60, 0.67].
[23]	2020	Greece	1	January 2017–December 2018	ICU patients in a public tertiary hospital	345	Invasive, respiratory, urinary, mucocutaneous, and wound infections	23,067 (binary, numerical, and categorical in total)	To compare the performance of eight ML algorithms to assess antibiotic susceptibility predictions	ML toolkit: WEKA—Data Mining Software in Java Workbench; LIBLINEAR LR and linear SVM; SVMs; SMO; instance-based learning (k-NN); J48; RF; RIPPER; MLP	TP rate, FP rate, precision, recall, F-measure, mmc, AUROC, precision-recall plot	The best performances were obtained with the RIPPER algorithm (F-measure of 0.678) and the MLP classifier (AUROC of 0.726).
[24]	2022	Taiwan	25	May 2013 to May 2014	Patients with healthcare-associated infections receiving at least one antimicrobial drug	7377	Healthcare-associated infection (bloodstream, urinary, pneumonia and surgical site infection).	26	To develop accurate and efficient ML models for auditing appropriate surgical antimicrobial prophylaxis	Supervised ML classifiers (Auto-WEKA (Bayesian optimisation method), MLP (artificial neural network), decision tree, SimpleLogistic (LogitBoost e CART algorithm), bagging, SMOTE and AdaBoost)	TP rate, TN rate, FP, FN, AUC, precision, specificity, sensitivity, weighted average for the multiclass model, execution time	The ML technique with the best performance metrics was the MLP, with a sensitivity of 0.967, specificity of 0.992, precision of 0.967, and AUC of 0.992.
[25]	2022	Israel	1	June 2007 to January 2019	Patients with UTI and wound infections from Maccabi Healthcare Services (MHS) with at least one record of a positive wound infection culture	140,349 UTI and 7365 wound infections.	UTI and wound	Not specified	To understand and predict the personal risk of treatment-induced gain of resistance	ML	Personal predicted risk	Choosing the antibiotic treatment with the minimal ML-predicted risk of emergence of resistance reduces the overall risk of emergence of resistance by 70% for UTIs and 74% for wound infections compared to the risk for physician-prescribed treatments.
[26]	2022	USA	1	2008 to 2019	Patients who received intravenous antibiotic treatment for a duration between 1 and 21 days during an ICU stay, at Beth Israel Deaconess Medical Centre, Boston	18,988 (22,845 unique stays)	Respiratory (pneumonia) and UTI	43	To estimate patients’ ICU LOS and mortality outcomes for any given day under the alternative scenarios of if they were to stop vs. continue antibiotic treatment	AI-based CDSS: recurrent neural network autoencoder and a synthetic control-based approach. It uses a bidirectional LSTM autoencoder; PyTorch was used to create a bidirectional LSTM RNN	Patients’ ICU LOS (days, mean delta, root mean squared error), mortality outcomes, to stop vs. continue ATB treatment (mean days reduction); day(s), mean delta (days, *p*-value), MAPE, MAE, RMSE, AUROC	The model reliably estimates patient outcomes under the contrasting scenarios of stopping or continuing ATB treatment: impact days where the potential effect of the unobserved scenario was assessed showed that stopping ATB therapy earlier had a statistically significant shorter LOS (mean reduction 2.71 days, *p*-value < 0.01). No impact on mortality was observed.
[27]	2021	Greece	1	January to December 2018	Patients admitted to the internal medicine wards of a public hospital	499 patients (11,496 instances)	Not specified	6 (attributes of sex, age, sample type, Gram stain, 44 antimicrobial substances, and the antibiotic susceptibility results)	To assess the effectiveness of AutoML-trained models to predict AMR	AutoML techniques using Microsoft Azure AutoML; SMOTE; algorithms: StackEnsemble, VotingEnsemble, MaxAbsScaler, LightGBM, SparseNormalizer, XGBoostClassifier	AUROC, AUCW, APSW, F1W, and ACC	The stack ensemble technique achieved the best results in the original and balanced dataset, with an AUCW metric of 0.822 and 0.850, respectively.
[28]	2020	EUA	1	December 2015 to August 2017	Patients hospitalized who received at least one antimicrobial from a list of those routinely tracked by the ASP at University of California, San Francisco Medical Centre	9651	Bloodstream, UTI, etc.	More than 200	To predict whether antibiotic therapy required stewardship intervention on any given day compared to the criterion standard of note left by the antimicrobial stewardship team in the patient’s chart	LR and boosted tree models	AUROC, Brier score, sensitivity, specificity, PPV, and NPV	Logistic regression and boosted tree models had AUROCs of 0.73 (95% CI, 0.69–0.77) and 0.75 (95% CI, 0.72–0.79) (*p* = 0.07), respectively.
[29]	2020	Israel	1	2001 to 2012	ICU adult patients are patients suspected of having a community-acquired bacterial infection	10,290 patients (12,232 ICU encounters)	Non-specified bacterial infection	Not specified	To identify ICU patients with low risk of bacterial infection as candidates for earlier EAT discontinuation	ML algorithms, including ridge regression, RF, SVC, XG Boost, K- NN, and MLP	AUROC, NPV, F1, precision, recall, high sensitivity threshold, TN, FP, FN, TP	Using structured longitudinal data collected up to 24, 48, and 72 h after starting EAT, the best models identified patients at low risk of bacterial infections with AUROCs up to 0.8 and negative predictive values > 93%. The T = 24 h RF model was the best performing model within this timepoint: AUC of 0.774, F1 of 0.424, NPV of 0.944, precision of 0.277, recall of 0.905, high sensitivity threshold of 0.258.
[30]	2019	USA	27	October 2015 to September 2017	Patients from the Duke Antimicrobial Stewardship Outreach Network (DASON) (Duke University School of Medicine)	382,943	Not specified	More than 100 features, including demographic data, length of stay, comorbidity, etc.	To identify patient- and facility-level predictors of antimicrobial usage in hospitalized patients using an ML approach, which can be used to inform a risk adjustment model to facilitate assessment of antimicrobial utilization	SVR and CB models	Root-mean-square error values	Both the SVR and CB models show better predictive accuracy than the null LM and null NB-GLM models (null statistical models) for all SAAR (external comparator) groups. CB performed better than SVR, according to the RMSE values (5.51 vs. 7.17 for all antibiotics, respectively).
[31]	2021	USA	3	October 2015 to September 2017	Adult and pediatric inpatient from Duke University Health System	170,294	Not specified	204	To evaluate whether variables derived from the electronic health records accurately identify inpatient antimicrobial use	A 2-stage RF ML modeling	AUROC and absolute error	Models accurately identified antimicrobial exposure in the testing dataset: the majority of AUCs were above 0.8, with a mean AUC of 0.85.
[32]	2022	USA	1	July 2017 to December 2019	Patient with antimicrobial orders from University of Maryland Medical Centre	17,503	Sepsis/bacteremia, bone/joint, central nervous system, cardiac/vascular, gastrointestinal genitourinary, respiratory, nonsurgical prophylaxis, skin and soft tissue infection, mycobacterial infection, neutropenia, surgical prophylaxis	33	To understand which patient and treatment characteristics are associated with either a higher or lower likelihood of intervention in a PAF program and to develop prediction models to identify antimicrobial orders that may be safely excluded from the review	LR, RF	Sensitivity, specificity, C-statistic, the out-of-bag error rate	The RF model had a C-statistic of 0.76 (95% CI, 0.75–0.77), with a sensitivity and specificity of 78% and 58%, respectively. This model would reduce review caseloads by 49%.
[33]	2018	Italy	1	March 2012 to 2019	Patients with nosocomial (UTI) from Principe di Piemonte Hospital in Senigallia	1486	UTI	6 (5 predictors + MDR resistance)	To design, develop, and evaluate, with a real antibiotic stewardship dataset, a predictive model useful for predicting MDR UTI onset after patient hospitalization	Catboost, support vector machine, and NN	Accuracy, AUROC, AUC-PRC, F1 score, sensitivity, specificity, MCC. FP, FN, TP, and TN	The ML method catboost had the best predictive results (MCC of 0.909; sensitivity of 0.904; F1 score of 0.809; AUC-PRC of 0.853, AUROC of 0.739; ACC of 0.717).
[34]	2020	Singapore	1	June 2016 to November 2018	Patients with uncomplicated URTI at the emergency department at Tan Tock Seng Hospital	715	Upper respiratory tract infections	50 (univariate analysis), 8 included in the algorithm	To develop prediction models based on local clinical and laboratory data to guide antibiotic prescribing for adult patients with uncomplicated upper respiratory tract infections	LR models, LASSO, and CART	AUC, sensitivity, specificity, PPV, NPV	The AUC on the validation set for the models was similar: LASSO: 0.70 [95% CI: 0.62–0.77], LR: 0.72 [95% CI: 0.65–0.79], decision tree: 0.67 [95% CI: 0.59–0.74].
[35]	2020	USA	2	2007 to 2016	Patients presenting with uncomplicated UTI at Massachusetts General Hospital and the Brigham and Women’s Hospital in Boston	10,053 (training dataset); 3629 (test set)	UTI	8	To predict antibiotic susceptibility using electronic health record data and build a decision algorithm for recommending the narrowest possible antibiotic to which a specimen is susceptible	LR, decision tree, and RF models	AUROC, FN rates	Decision trees and RF were excluded based on their poor validation set performance and relative lack of interpretability. The LR model provided antibiotic stewardship for a common infectious syndrome by maximizing reductions in broad-spectrum antibiotic use while maintaining optimal treatment outcomes. The algorithm achieved a 67% reduction in the use of second-line antibiotics relative to clinicians and reduced inappropriate antibiotic therapy by 18%, close to the rate of clinicians.
[36]	2019	Cambodia	1	February 2013 to January 2016	Children with at least one positive blood culture from Angkor Hospital for Children	195 (training set); 48 (model validation)	Bloodstream	35	To predict Gram stains and whether bacterial pathogens could be treated with standard empiric antibiotic regimens	RF, LR, decision trees constructed via recursive partitioning, boosted decision trees using adaptive boosting, linear SVM, polynomial SVM, radial SVM, and k-NN	AUROC	The RF method had the best predictive performance overall: AUC of 0.80 (95% CI 0.66–0.94) for predicting susceptibility to ceftriaxone, 0.74 (0.59–0.89) for susceptibility to ampicillin and gentamicin, 0.85 (0.70–1.00) for susceptibility to neither, and 0.71 (0.57–0.86) for Gram stain result.
[37]	2022	Italy	2	January 2012 to December 2020	Women affected by recurrent UTI who had undergone antimicrobial treatment for uncomplicated lower UTI	1043	Recurrent UTI	Not specified	To define an NN for predicting the clinical and microbiological efficacy of antimicrobial treatment of a large cohort of women affected by recurrent UTIs for use in everyday clinical practice	NN	Sensitivity, specificity, HR	The use of artificial NN in women with recurrent cystitis showed a sensitivity of 87.8% and specificity of 97.3% in predicting the clinical and microbiological efficacy of the prescribed antimicrobial treatment.

Notes: APSS—antimicrobial prescription surveillance system; ICU—intensive care unit; CI—confidence interval; CDSS—clinical decision support system; UTI—urinary tract infection; RESSEL—reliable semi-supervised ensemble learning; RF—random forest; PPV—positive predictor value; NPV—negative predictive value; AUC—area under the curve; AUROC—area under the ROC curve; Nneg—the number of UTI-negative labeled cultures; Npos—the number of UTI-positive labelled cultures; N—number; Ntrain—number in training set; Nval—number in validation set; Ntest—number in test set; TP—true positive; TN—true negative; FP—false positive; FN—false negative; Mmc—a correlation coefficient; LOS—length of stay; MAPE—mean absolute; MAE—mean absolute error; RMSE—root-mean-squared error; AUCW—area under the curve-weighted; APSW—average precision score-weighted; F1W—F1 score-weighted; ACC—accuracy; ML—machine learning; LR—logistic regression; SVM—support vector machine; XGB—eXtreme Gradient Boosting; NN—nearest neighbors; SMO—sequential minimal optimization; MLP—multilayer perceptron; LSTM—long short-term memory; RNN—recurrent neural network; ATB—antibiotic; AutoML—automated machine learning; SMOTE—synthetic minority oversampling technique; EAT—empiric antibiotic therapy; SVC—support vector classifier; SVR—support vector regression; CB—cubist regression; LM—linear model; null NB-GLM model—negative binomial generalized linear model; SAAR—standardized antimicrobial administration ratio; PAF—prospective audit with feedback; MDR—multidrug resistant; AUC-PRC—area under precision recall curve; MCC—Matthews correlation coefficient; HR—hazard ratio. Scikit-learn package—available at https://scikit-learn.org/stable/ (accessed on 24 March 2024). WEKA—Data Mining Software—WEKA 3.6. Auto-WEKA—2.0. Pytorch—https://pytorch.org/ (accessed on 24 March 2024). Microsoft Azure AutoML—https://learn.microsoft.com/en-us/azure/?product=popular (accessed on 24 March 2024). SMOTE—https://learn.microsoft.com/en-us/azure/machine-learning/component-reference/smote?view=azureml-api-2 (accessed on 24 March 2024).

**Table 2 antibiotics-13-00307-t002:** Characteristics of the features of the included studies.

Study	Demographics	Adult	Paediatric	Clinical	Laboratory/Microbiological	Comorbidities	Type of Infection	ICU
[20]	Yes	Yes	No	Yes	Yes	No	No	Yes
[21]	Yes	Yes	Yes	Yes	Yes	Yes	Yes	No
[22]	Yes	Yes	No	Yes	Yes	Yes	Yes	No
[23]	Yes	Yes	No	No	Yes	No	Yes	Yes
[24]	Yes	Yes	Yes	Yes	No	No	Yes	Yes
[25]	Yes	Yes	No	Yes	Yes	Yes	Yes	Not specified
[26]	Yes	Yes	Not specified	Yes	Yes	No	No	Yes (only ICU patients)
[27]	Yes	Yes	No	No	Yes	No	Yes	No
[28]	Yes	Yes	No	Yes	Yes	No	Yes	Yes
[29]	Yes	Yes	No	Yes	Yes	Yes	Yes	Yes (only ICU patients)
[30]	Yes	Yes	No	Yes	Yes	Yes	Yes	Not specified
[31]	Yes	Yes	Yes	Yes	Yes	Yes	Not specified	Yes
[32]	Yes	Yes	No	Yes	Yes	No	Yes	Not specified
[33]	Yes	Yes	Not specified	No	Yes	No	Yes	Not specified
[34]	Yes	Yes	No	Yes	Yes	Yes	Yes	No
[35]	Yes	Yes	No	Yes	Yes	Yes	Yes	Yes
[36]	Yes	No	Yes	Yes	Yes	No	Yes	Yes
[37]	Yes	Yes	No	Yes	Yes	Not specified	Yes	Not specified

Note: ICU—intensive care unit.

**Table 3 antibiotics-13-00307-t003:** Risk of bias assessment of the included studies by NIH Quality Assessment Tool for Observational Cohort and Cross-Sectional Studies.

Criteria\ Study	1	2	3	4	5	6	7	8	9	10	11	12	13	14	Quality Rating
[20]	Yes	Yes	NA	Yes	NR	Yes	Yes	NA	Yes	Yes	Yes	NR	NA	NR	Fair (57.1%)
[21]	Yes	Yes	NA	Yes	NR	Yes	Yes	NA	Yes	Yes	Yes	NR	NA	No	Fair (57.1%)
[22]	Yes	Yes	NA	Yes	NR	Yes	Yes	NA	Yes	Yes	Yes	NR	NA	Yes	Fair (64.3%)
[23]	Yes	Yes	NA	Yes	NR	Yes	Yes	NA	Yes	Yes	Yes	NR	NA	No	Fair (57.1%)
[24]	Yes	Yes	NA	Yes	NR	Yes	Yes	NA	Yes	Yes	Yes	NR	NA	NR	Fair (57.1%)
[25]	Yes	Yes	NA	Yes	NR	Yes	Yes	NA	Yes	Yes	Yes	NR	NA	NR	Fair (57.1%)
[26]	Yes	Yes	NA	Yes	NR	Yes	Yes	NA	Yes	Yes	Yes	NR	NA	Yes	Fair (64.3%)
[27]	Yes	Yes	NA	Yes	NR	Yes	Yes	NA	Yes	Yes	Yes	NR	NA	NR	Fair (57.1%)
[28]	Yes	Yes	NA	Yes	NR	Yes	Yes	NA	Yes	Yes	Yes	NR	NA	NR	Fair (57.1%)
[29]	Yes	Yes	NA	Yes	NR	Yes	Yes	NA	Yes	Yes	Yes	NR	NA	NR	Fair (57.1%)
[30]	Yes	Yes	NA	Yes	NR	Yes	Yes	NA	Yes	Yes	Yes	NR	NA	NR	Fair (57.1%)
[31]	Yes	Yes	NA	Yes	NR	Yes	Yes	NA	Yes	Yes	Yes	NR	NA	NR	Fair (57.1%)
[32]	Yes	Yes	NA	Yes	NR	Yes	Yes	NA	Yes	Yes	Yes	NR	NA	Yes	Fair (64.3%)
[33]	Yes	Yes	NA	Yes	NR	Yes	Yes	NA	Yes	Yes	Yes	NR	NA	NR	Fair (57.1%)
[34]	Yes	Yes	NA	Yes	NR	Yes	Yes	NA	Yes	Yes	Yes	NR	NA	NR	Fair (57.1%)
[35]	Yes	Yes	NA	Yes	Yes	Yes	Yes	NA	Yes	Yes	Yes	NR	NA	No	Fair (64.3%)
[36]	Yes	Yes	NA	Yes	NR	Yes	Yes	NA	Yes	Yes	Yes	NR	NA	NR	Fair (57.1%)
[37]	Yes	Yes	NA	Yes	NR	Yes	Yes	NA	Yes	Yes	Yes	NR	NA	NR	Fair (57.1%)

Notes: NA—not applicable; NR—not reported.

**Table 4 antibiotics-13-00307-t004:** Risk of bias and applicability assessment by PROBAST.

Study	Risk of Bias				Applicability			Overall	
	1. Participants	2. Predictors	3. Outcome	4. Analysis	1. Participants	2. Predictors	3. Outcome	Risk of Bias	Applicability
[20]	-	+	+	?	?	?	+	-	?
[21]	+	?	+	?	+	+	+	?	+
[22]	+	?	+	+	+	+	+	?	+
[23]	+	+	+	?	+	+	+	?	+
[24]	+	+	+	+	+	+	+	+	+
[25]	?	?	+	?	?	?	+	?	?
[26]	+	+	+	?	+	+	+	?	+
[27]	-	+	+	-	-	+	+	-	-
[28]	+	+	+	-	+	?	+	-	?
[29]	+	?	+	?	+	?	+	?	?
[30]	+	+	+	-	+	?	+	-	?
[31]	+	+	+	-	+	+	?	-	?
[32]	+	+	+	-	+	+	+	-	+
[33]	+	+	+	-	+	?	+	-	?
[34]	+	+	+	-	+	+	+	-	+
[35]	+	+	?	-	+	+	?	-	?
[36]	+	+	?	-	+	+	?	-	?
[37]	+	+	+	-	+	+	?	-	?

Notes: + low concern, - high concern, ? unclear.

## Data Availability

No new data were created or analyzed in this study. Data sharing is not applicable to this article.

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
