# Peer review of "Brave New World of Artificial Intelligence: Its Use in Antimicrobial Stewardship—A Systematic Review"

_antibiotics, 2024, doi:10.3390/antibiotics13040307_

Round 1

Reviewer 1 Report

Comments and Suggestions for Authors

Thank you for a great review. This review will be a foundation for future research in AI, AMR, and AMU relationships. From the review, is it possible to add some graphical presentation of the AUC, Se, Sp, and other performance metrics?

I know you did not plan to do a meta-analysis, I am just wondering if that can improve the quality of the review further since you just have 18 articles included in the review.

Author Response

Dear Reviewer,

Dear Reviewer,

We would like to thank you for the valuable comments, suggestions and the thoughtful evaluation of the manuscript. We have directly answered to the questions, and we hope that the manuscript has improved and is now ready for acceptance at “Antibiotics”. Changes in the revised manuscript have been marked up using the “track changes” function.

Comments and Suggestions for Authors

Thank you for a great review. This review will be a foundation for future research in AI, AMR, and AMU relationships. From the review, is it possible to add some graphical presentation of the AUC, Se, Sp, and other performance metrics?

I know you did not plan to do a meta-analysis, I am just wondering if that can improve the quality of the review further since you just have 18 articles included in the review.

R: The authors would like to thank the Reviewer for the comments. The authors agree that adding graphical presentations of the performance metrics would enrich the review. However, the authors believe that, due to the heterogeneity of the included studies, the graphical information could be biased. For this reason, the authors opted to present performance data descriptively.

Reviewer 2 Report

Comments and Suggestions for Authors

This review article aims to determine the usefulness of artificial intelligence in the use of Antimicrobial Stewardship that helps scientists and practitioners consider to use of artificial intelligence in the prevention and treatment of infectious diseases. Therefore, this review is worth to be considered for publication after revision in line with the comments given below:

1.      Studies regarding exclusively antibiotic resistance or outpatient populations were excluded. Wouldn’t this data provide the usefulness of AI in understanding antibiotic resistance risks?

2.      All studies used in this review have suggested ML algorithms' usefulness in assisting antimicrobial stewardship teams in multiple tasks, such as identifying inappropriate prescribing practices, choosing the appropriate antibiotic therapy, or predicting AMR. However, a more general benefit of AI is given in the conclusion but not a more specific outcome/s of the results of these studies. At the end of this review article, one cannot understand if ML algorithms can be applied to assist antimicrobial stewardship teams in multiple tasks, such as identifying inappropriate prescribing practices, choosing the appropriate antibiotic therapy, or predicting AMR.

3.      “The features included in the algorithms were divided into the following groups: demographics, adult patients, pediatric patients, clinical, laboratory/ microbiological, comorbidities, type of infection, and ICU”. What do your evaluation results of this review suggest about the usefulness of AI in use of for each of these groups?

4.      In the subtitle “Limitations of the studies included” it has been regarded as antifungal or antiviral use and the exclusion of studies focusing on HIV, parasitic diseases, or tuberculosis as the limits of this review. Most of the references reviewed in this manuscript are related to antibiotic use so it might be more appropriate to narrow the content of this review to bacterial infections so that there would not be a limitation in this case.

5. Table 1 is very large. It could be considered to summarize its content to make a more concise presentation.

Dear Editor,

This manuscript reviews the use of artificial intelligence to prevent and treatment of infectious diseases. All papers related to artificial intelligence usage in infectious diseases were selected and summarized to determine the potential for their prevention and treatment. The presentation and evaluation of the published research articles were reviewed and presented well in general.  Therefore, I suggest that this review be considered for publication in the Antibiotics journal after revision following the comments made. 

Kind regards.

Author Response

Dear Reviewer,

We would like to thank you for the valuable comments, suggestions and the thoughtful evaluation of the manuscript. We have directly answered to the questions, and we hope that the manuscript has improved and is now ready for acceptance at “Antibiotics”. Changes in the new manuscript have been marked up using the “track changes” function.

Comments and Suggestions for Authors

This review article aims to determine the usefulness of artificial intelligence in the use of Antimicrobial Stewardship that helps scientists and practitioners consider to use of artificial intelligence in the prevention and treatment of infectious diseases. Therefore, this review is worth to be considered for publication after revision in line with the comments given below:

  1. Studies regarding exclusively antibiotic resistance or outpatient populations were excluded. Wouldn’t this data provide the usefulness of AI in understanding antibiotic resistance risks?

R: The authors would like to thank the Reviewer for the comments. In fact, within the search strategy used, the authors found many studies regarding antibiotic resistance. However, those studies applied AI in the detection mechanisms of antimicrobial resistance or evaluation of the impact of antimicrobial susceptibility testing, which fall out of the scope of the systematic review. Most studies in the outpatient population lacked important information regarding the features of machine learning methods or algorithms. For these reasons, only one study regarding the outpatients population assisted in a General hospital was included (study no. 41).

  1. All studies used in this review have suggested ML algorithms' usefulness in assisting antimicrobial stewardship teams in multiple tasks, such as identifying inappropriate prescribing practices, choosing the appropriate antibiotic therapy, or predicting AMR. However, a more general benefit of AI is given in the conclusion but not a more specific outcome/s of the results of these studies. At the end of this review article, one cannot understand if ML algorithms can be applied to assist antimicrobial stewardship teams in multiple tasks, such as identifying inappropriate prescribing practices, choosing the appropriate antibiotic therapy, or predicting AMR.

R: Information was added in the conclusion section (please see line 450-452).

  1. “The features included in the algorithms were divided into the following groups: demographics, adult patients, pediatric patients, clinical, laboratory/ microbiological, comorbidities, type of infection, and ICU”. What do your evaluation results of this review suggest about the usefulness of AI in use of for each of these groups?

R: The authors intended to evaluate which type of features were used in the ML algorithms of the selected studies. The authors believe that different types of features should be included in the algorithms to strengthen the prediction model and help clinical decisions. This information is explained in lines 303-307 as well as in lines 450-452.

4. In the subtitle “Limitations of the studies included” it has been regarded as antifungal or antiviral use and the exclusion of studies focusing on HIV, parasitic diseases, or tuberculosis as the limits of this review. Most of the references reviewed in this manuscript are related to antibiotic use so it might be more appropriate to narrow the content of this review to bacterial infections so that there would not be a limitation in this case.

R. The authors total agree with the Review suggestion. This information has been deleted and “studies not focusing on bacterial infections” were added to the exclusion criteria. References 49, 50 and 51 were deleted.

  1. Table 1 is very large. It could be considered to summarize its content to make a more concise presentation.

R. We have tried to summarize all the main results from the studies. However, Table 1 contains important information indispensable for the reader understanding of the included studies of the systematic review.

Reviewer 3 Report

Comments and Suggestions for Authors

This is a publications review of artificial intelligence (AI) on support of antimicrobial stewardschip. By selecting publications on this theme, the authors analyse the results of 18 manuscripts.

It is a domain of research in rapid extension.

The manuscript put in evidence the very restricted help of artificial intelligence in antimicrobial stewardship (limited in number of relevant publications; limited in scope and heterogeneity of infectious diseases covered; limited in homogeneity of methodological approach).

These limitations are not clearly expressed in the manuscript, even if they are evident. For "old" practitionners, "AI" looks like a "mirage", an "illusion", or even a "delusion". The authors should at least discuss this perception by many practitioners, and I am sure that many "young" practitionners share the same opinion without daring to say so (social pressure for new electronic tools). So, a more detailed discussion on the benefit of AI in real life for the next future is welcome. Of course, expressing the opinion of the authors. What is your practical change in antimicrobial stewardship after this review? Will you support major investments in this approach whithin your hospitals (electronic material, data analysts)? How many data analysts? Material cost? Cost/benefit analysis?

Editing: it is surprising that the numerotation of the "studies" corresponds actually to the numerotations of the "reference". It should be more clear to numerotate the studies from 1 to 18 as following: "1 (ref 20); 2 (ref 27); 2 (ref 28); ...; 18 (ref 43)

Author Response

Dear Reviewer,

We would like to thank you for the valuable comments, suggestions and the thoughtful evaluation of the manuscript. We have directly answered to the questions, and we hope that the manuscript has improved and is now ready for acceptance at “Antibiotics”. Changes in the new manuscript have been marked up using the “track changes” function.

Comments and Suggestions for Authors

This is a publications review of artificial intelligence (AI) on support of antimicrobial stewardschip. By selecting publications on this theme, the authors analyse the results of 18 manuscripts.

It is a domain of research in rapid extension.

The manuscript put in evidence the very restricted help of artificial intelligence in antimicrobial stewardship (limited in number of relevant publications; limited in scope and heterogeneity of infectious diseases covered; limited in homogeneity of methodological approach).

These limitations are not clearly expressed in the manuscript, even if they are evident. For "old" practitionners, "AI" looks like a "mirage", an "illusion", or even a "delusion". The authors should at least discuss this perception by many practitioners, and I am sure that many "young" practitionners share the same opinion without daring to say so (social pressure for new electronic tools). So, a more detailed discussion on the benefit of AI in real life for the next future is welcome. Of course, expressing the opinion of the authors. What is your practical change in antimicrobial stewardship after this review? Will you support major investments in this approach whithin your hospitals (electronic material, data analysts)? How many data analysts? Material cost? Cost/benefit analysis?

R. The authors would like to thank the Reviewer for the comments. The authors opinion have been added to the discussion (please see line 323 to 330)

Editing: it is surprising that the numerotation of the "studies" corresponds actually to the numerotations of the "reference". It should be more clear to numerotate the studies from 1 to 18 as following: "1 (ref 20); 2 (ref 27); 2 (ref 28); ...; 18 (ref 43)

R. We intentionally referenced the studies at the beginning of the results so that the reference would coincide with the number on the tables.